# Health-Related Quality of Life (HRQOL) Instruments and Mobility: A Systematic Review

**DOI:** 10.3390/ijerph192416493

**Published:** 2022-12-08

**Authors:** Natalia Hernández-Segura, Alba Marcos-Delgado, Arrate Pinto-Carral, Tania Fernández-Villa, Antonio J. Molina

**Affiliations:** 1Institute of Biomedicine (IBIOMED), University of León, 24071 Leon, Spain; 2Department of Nursing and Physiotherapy, SALBIS Research Group, Campus de Ponferrada, Universidad de León, Avda/Astorga, s/n, 24071 Leon, Spain

**Keywords:** quality of life, mobility, movement, International Classification of Functioning, Disability and Health (ICF), measurement instruments, generic instruments

## Abstract

Physical function is one of the most important constructs assessed in health-related quality of life (HRQOL), and it could be very useful to assess movement ability from the perspective of the patient. The objective of this study was to compare the content of the domains related to mobility covered by the HRQOL questionnaires based on the International Classification of Functioning, Disability and Health (ICF) and to evaluate their quality according to the COnsensus-based Standards for the selection of health Measurement INstruments (COSMIN) guidance. For this, a systematic review was carried out in the databases Scopus, Web of Science and Science Direct. The inclusion criteria were development and/or validation studies about generic HRQOL measures, and the instruments had to include items related to mobility and studies written in English or Spanish. The comparison of content was performed using the ICF coding system. A total of 3614 articles were found, 20 generic HRQOL instruments were identified and 120 (22.4%) mobility-related items were found. Walking was the most represented category. Low-quality evidence on some measurement properties of the generic HRQOL instruments was revealed. The CAT-Health is a useful questionnaire to be used in rehabilitation due to its psychometric properties and its content.

## 1. Introduction

Health status measures are widely used in both research and clinical practice, including technical, clinical and patient-oriented measures. These measures are useful for determining patients’ problems, management and evaluation of the effect of an intervention [1,2], so their use is essential to improve the quality of healthcare [3,4].

Patient-oriented measures collect information that comes directly from the patient [5]. These measures, which are also called patient-reported outcome measures (PROMs), include patient satisfaction, community integration and social participation [3] from the patient’s perspective, among others. This perspective of the patient is especially important in research and care practice, specifically in functioning and health, so concepts and instruments to measure them, such as quality of life, health-related quality of life (HRQOL) or health status, are needed [6]. In rehabilitation and physiotherapy, it could be very useful to assess movement capacity from the patient’s perspective using this type of instrument, and not only from biological function [7], since the physical function is one of the most important constructs to be assessed.

In this regard, it is interesting to consider the link between HRQOL and physical function since, although there is no total consensus on the definition of the concept of HRQOL [8,9], most HRQOL concepts and instruments include items related to the physical dimension of the individual, in addition to others related to the mental and social dimensions. Such HRQOL assessment instruments can be differentiated into two major kinds, specific or generic, depending on whether they purport to be applicable to specific patient populations or diagnostic groups or whether they are designed to be broadly applicable across types and severities of disease and to all demographic and cultural subgroups [10].

Most of these generic HRQOL instruments cover the three dimensions [11], but with varying percentages in the number of items that each instrument allocates to each dimension and with different approaches to the idea of physical function and mobility [6,12]. It is relevant to know to what extent and with what orientation the HRQOL instruments assess the different dimensions and especially the physical dimension, in order to know more deeply their direct usefulness in practice and the expected correlation with other assessments of physical function.

The universal conceptual framework for functioning and health, provided by the International Classification of Functioning, Disability and Health (ICF) [13], allows comparison between different instruments [1,14]. Thus according to other studies [6,15,16], a comparison based on the ICF can provide information on which instrument could be the most useful in the clinic of rehabilitation patients, depending on the type of patient, the length of the instrument, the answer options and the psychometric properties.

To date, several reviews have been published comparing the ICF content of specific HRQOL instruments [12,17,18,19,20], but only one article has been found comparing generic questionnaires [6]. In this article, Cieza and Stucki select only six instruments to relate to ICF. For this reason, it is of interest to carry out a comparison and evaluation of the content of all the generic HRQOL instruments to facilitate their comparability and allow the selection of the instrument that would be most useful in clinical practice.

The objectives of our study were (1) to identify generic HRQOL measures that include a domain related to mobility or physical function, (2) to assess their quality and (3) to compare the content of the ICF-based mobility-related domains covered by the HRQOL questionnaires.

## 2. Materials and Methods

We carried out a systematic review following the COnsensus-based Standards for the selection of health Measurement INstruments (COSMIN) guidance [21] and the Preferred Reporting Items for Systematic Reviews and Meta-Analyses (PRISMA) statement [22]. The protocol was registered in the prospective international register of systematic reviews PROSPERO with ID CRD42020176035.

### 2.1. Data Sources and Searches

In March 2020, a search was conducted in the following databases: Science Direct, Scopus and Web of Science (which includes Medline, Current Contents Connect, Derwent Innovations Index, KCI-Korean Journal Database, Russian Science Citation Index, SciELO Citation Index, and the main Web of Science collection). The search terms were “health related quality of life”, mobility, function, physical, scale, questionnaire, survey, test, instrument, index, psychometrics, validity and reliability; all details are shown in Table 1. The online tool Parsifal was used to detect duplicates.

### 2.2. Study Selection

The inclusion criteria were (1) development and/or validation studies for PROMs about HRQOL and (2) the instruments should include mobility-related items. The included studies were further restricted to those on adults without any specific pathology. No restrictions were imposed regarding the date of publication, although only studies written in English or Spanish were selected. 

The exclusion criteria were (1) seniors, children or adolescents or both; (2) experimental, analytical, descriptive and review; and (3) translations or cross-cultural validations of instruments studies. After the removal of duplicate studies, two independent reviewers (NHS and AMD) assessed all titles and abstracts. Consensus on inclusion was sought between reviewers in a meeting, and, in case of disagreement, a third reviewer (AJM) arbitrated.

### 2.3. Data Extraction

For data extraction, a form was drawn up in an excel spreadsheet in which a reviewer (NHS) recorded data on the measurement instrument name, author of the original instrument and year of publication, domains included in the instrument, answer choices, the number of items, the number of items on mobility and the ICF domains explored. Moreover, the following data were extracted from each publication: author, year of publication, study sample, administration method, country and language. The data were checked by a second reviewer (AMD).

### 2.4. Linkage of Items to the ICF

We applied rules that were described for linking the concepts about mobility contained in the HRQOL measures to the ICF [1,2,23]. If an item of a measure contained more than one concept, each concept was linked separately. Only those dimensions that included mobility-related items were tied. The response options of an item were bound to the ICF if they contained meaningful concepts, which are concepts that could be linked to the ICF. If a meaningful concept of an item was explained by examples, both the concept and examples were linked. Meaningful concepts referring to health or physical health, in general, were assigned “nd-gh” or “nd-ph” (not definable-general health, not definable-physical health), respectively, and those referring to the quality of life, in general, were assigned “nd-qol” (not definable-quality of life). If the meaningful concept refers to a diagnosis or a health condition was assigned “hc” (health condition), and if the meaningful concept is not contained in the ICF, “nc” (not covered by ICF) was assigned.

### 2.5. Quality Assessment

In order to assess the quality of studies, two researchers (NHS and AMD) independently rated the methodological quality of the eligible studies using the COSMIN checklist [24,25,26]. The following measurement properties were assessed: content validity, structural validity, internal consistency, reliability, measurement error, criterion validity and construct validity. Responsiveness was not assessed due to the selection instruments validated in the general population, and cross-cultural validity due to translations or cross-cultural adaptations studies were excluded. Only the eligible studies and the instrument development was reviewed. In addition, the COSMIN website was checked to see if the quality of the PROM development was already rated in another review.

First, each study was rated according to a four-point rating scale as very good, adequate, doubtful or inadequate quality [24,25,26]. Second, this rating was used to obtain a separate score for each measurement property and the instrument development. As per the COSMIN methodology, the reported worst score counts. Third, each criterion could be rated as sufficient (+), insufficient (-) or indeterminate (?) [24]. Lastly, the results of all studies, each instrument and the reviewer´s rating were summarized qualitatively as sufficient (+) if the ratings per study were all sufficient, insufficient (-) if each rating was insufficient or an inconsistent (±) rating. Content validity was scored on the relevance, comprehensibility and comprehensiveness of the PROM to a general population.

The quality of evidence was rated according to the Modified Grading of Recommendations, Assessment, and Evaluation (GRADE) approach [24] into high (the true measurement property lies close to that of the estimate of the measurement property), moderate (the true measurement property is likely to be close to the estimate of the measurement property), low (the true measurement property may be substantially different from the estimate of the measurement property) or very low (the true measurement property is likely to be substantially different from the estimate of the measurement property).

## 3. Results

The searches identified 3614 records obtained in Science Direct (376), Scopus (1762) and Web of Science (1476). A total of 1222 duplicates were eliminated via Parsifal. The title and abstract were reviewed, and 2237 publications were identified that did not meet the inclusion criteria and were excluded. From 151 records remaining, full-text articles were retrieved, and 136 were excluded: 65 did not include mobility-related items in their instruments, 52 were translations into languages other than English or Spanish or cross-cultural adaptations, 21 assessed measurement properties in the elderly and 4 studies were in patients with specific pathology (headache, back pain, cardiovascular diseases and disorders of the knee). Some studies had more than one reason for being excluded. Finally, 20 studies were included (Figure 1) [27,28,29,30,31,32,33,34,35,36,37,38,39,40,41,42,43,44,45,46].

Twenty generic HRQOL instruments were identified from the selected studies (Table 2). Most of them (19 instruments) were multidimensional, and all the instruments were self-administered. Given the inclusion criteria, the 20 PROMs had items related to mobility or movement; in total, 120 mobility-related items were found from 535 items. Most questionnaires offered between three and six response options (Likert scale) or a dichotomous answer (yes/no). Two instruments offered response options in visual analog scale (VAS) format (Table 2). Table 3 shows the characteristics of the samples of each study and information on the instrument administration.

### 3.1. Linking to ICF

The results for quantification of the ICF categories measured by the instruments are shown in Table 4, Table 5 and Table 6. The numbers contained in the tables show the frequency with which each ICF category was addressed in the instruments. A total of 285 concepts were linked to ICF from 120 items. *Body functions* were covered by five instruments (25%) (Table 4), *environmental factors* by four instruments (20%) (Table 5) and *activities and participation* by all examined instruments (Table 6).

Aspects of *mobility* were not represented in the SF-6D, in the HALex instruments and neither in the MQLI. SF-6D covered *activities and participation* in general, while HALex only included *domestic life* and two items linked with *major life areas*. Other categories from *activities and participation* were covered by the MQLI, which included *self-care* (Appendix A). *Walking*, which was contained in 10 instruments (50%), was the most represented category, followed by *climbing* and *dressing* (contained in nine instruments, 45%). With respect to *self-care* and *domestic life*, 11 and 12 instruments, respectively, covered some categories of these domains. SWED-QUAL and MQLI had two concepts not definable (nd), and FSQ 1, SF-36 and SF-12 had a concept that was not definable-general health (nd-gh). QWB-SA had two concepts that were nd-gh and two concepts that were not definable-physical health (nd-ph).

### 3.2. Quality Assessment

Six studies [36,40,45,47,50,51] reported information on the instrument’s development (see Table 7). Moreover, Table 7 includes the rating of COSMIN website reviews [53,54]. All instruments scored inadequate in this section. Only one [45] of these six articles assessed the content validity and no other measurement property.

The most frequently studied measurement properties were internal consistency (15 studies) and constructed validity (13 studies). Eight instruments had high-quality evidence for sufficient internal consistency, and six questionnaires had high-quality evidence for insufficient internal consistency; only MQLI had low methodological quality (Table 8). Given the inclusion criteria, all the studies were in the general population. Low-quality evidence for sufficient measurement error was found for all studies that include this property. All measurement properties assessed by NHP had high quality, but internal consistency was insufficient. Measurement properties results are shown in Table 8.

## 4. Discussion

This review revealed that mobility is an important construct within HRQOL, expending almost a quarter of the items in their assessment. Moreover, the ICF was a useful tool when comparing HRQOL instruments since its content was represented by the ICF categories, serving as a common conceptual framework. On the other hand, this systematic review highlights the low-quality evidence on some measurement properties, such as reliability or construct validity, of the generic HRQOL instruments.

The domains most frequently assessed were *mobility*, *self-care* and *domestic life*. A systematic review [59] with mobility instruments reported similar results. The percentage of items related to mobility or movement is variable in each questionnaire, asking most about the transfer of the body in space. A total of 22.4% of the items were related to mobility, ranging from 60.7% of the Stark QoL to 3.8% of the WHOQOL-BREF. However, the majority (62%) were between 17 and 28%. Cieza and Stucki [6] highlighted the scarce representation of mobility in the WHOQOL-BREF and in the EQ-5D, which only covers *mobility* in general and *walking*. For this reason, these instruments are not the most recommended for use in people with mobility problems.

According to the two-level classification, *walking* was the category more prevalent, followed by *moving around*, both related to the transfer of the body in space. Both categories were included in 10 instruments (50%). On the other hand, the Stark QoL and the SF-36 assess four concepts from the *carrying, moving and handling objects* category. Only the PAT-5D-QOL links more concepts in this category, but this instrument is a semi-adaptive questionnaire, so not all items are answered. That is the reason why these questionnaires can be useful to assess the HRQOL in people with dysfunctions of the upper extremity. Moreover, the Stark QoL includes pictures and can therefore be useful for assessing people with comprehension problems. Based on the third-level classification, *climbing* was the most represented category, followed by *dressing* and *walking*. The NHP and the FSQ include the three categories.

Otherwise, regardless of adaptive questionnaires, the QWB-SA is the most appropriate for assessing HRQOL in people with problems when changing or maintaining body positions. In patients in whom the transfer of the body in the space may be influencing their HRQOL, the FSQ or the SF-36 can be useful. In order to assess fine hand use, the HUI-3 or the SWED-QUAL should be selected. For all these reasons, and due to the assessment of the measurement properties, we recommend using the SF-36 in people with upper limb disabilities or with problems transferring the body in space. However, this questionnaire can overestimate HRQOL in patients with fine motor problems, so it would be convenient to develop a new instrument with better psychometric properties that include finding hand use.

The instruments that presented the highest methodological quality were CAT-Health [40,41] and SF-36 [35], although few properties were assessed. The SF-36 has 8 dimensions and 10 items related to mobility. The CAT-Health is a computerized test composed of 96 items, of which 25 are mobility-related. Being computerized, it can cover many categories of ICF, although only between 5 and 15 items are completed. Both instruments have been designed for the general population, although SF-36 has been validated in specific populations susceptible to rehabilitation as patients with low back pain or post-rib fracture [57,58] with lower methodological quality. It would also be interesting to validate the CAT-Health in rehabilitation patients.

On the other hand, the instrument with the most linked concepts is the PAT-5D-QoL, but as in the CAT-Health, not all items are answered since it is semi-adaptive, so it should be considered when comparing it with other questionnaires. The PAT-5D-QoL has 130 questions, 78 of them related to mobility, of which only 30 are answered, 18 related to mobility. This questionnaire can be an alternative for people who are not familiar with the use of technology. Future studies should develop and validate an HRQOL computerized adaptive test and a semi-adaptive paper version with adequate methodological quality. The adaptive test allows for reducing the duration of the test and that only the items paired with the ability level of the respondent are answered [60]. Therefore, creating a questionnaire of this type will make it possible to assess the HRQOL in people with different mobility limitations. 

This work has some limitations that should be taken into account. As with any review, the results obtained are subject to biases derived from the studies included in the review. However, the methodological quality analysis performed provides a good way to discriminate the usefulness and reliability of the instruments. Moreover, not all databases were searched, such as PsycINFO, EconLit and EMBASE. Another limitation of the study is the exclusion of the questionnaires validated in the elderly since including in the analysis the articles in which the validation was performed on the elderly could have given a different perspective on the importance of mobility in the quality of life. Finally, some properties assessed by other authors [61] not included in the COSMIN website were not included in this review. The recommendations of this guideline were followed since it is a current reference consensus document for the evaluation of PRO instruments, thus limiting the variability among researchers who assess psychometric properties.

Despite the limitations, our study also has several strengths. We highlight that, to the best of our knowledge, it is the first study that reviews the HRQOL instruments and links its content with the ICF. Throughout the systematic review, a strict methodology has been used to assess the quality and biases of the articles found. On the other hand, in addition to evaluating the content of the instruments, this work assesses the measurement properties.

## 5. Conclusions

Twenty generic HRQOL instruments that assess mobility were identified, and approximately one-quarter of their items were related to mobility in thirteen of these instruments. The most frequently assessed categories were walking and moving around, including in twelve and thirteen questionnaires, respectively, and eleven instruments had items related to upper limb activities. The adaptive tests were the measures with the most items related to mobility, but not all items were answered in this type of questionnaire. The instruments that presented the highest methodological quality were CAT-Health and SF-36. Both questionnaires assess the transfer of the body in space, the movement of the body position and upper extremity function. The CAT-Health is a useful questionnaire to be used in rehabilitation due to its psychometric properties, its content and its duration. 

## Figures and Tables

**Figure 1 ijerph-19-16493-f001:**
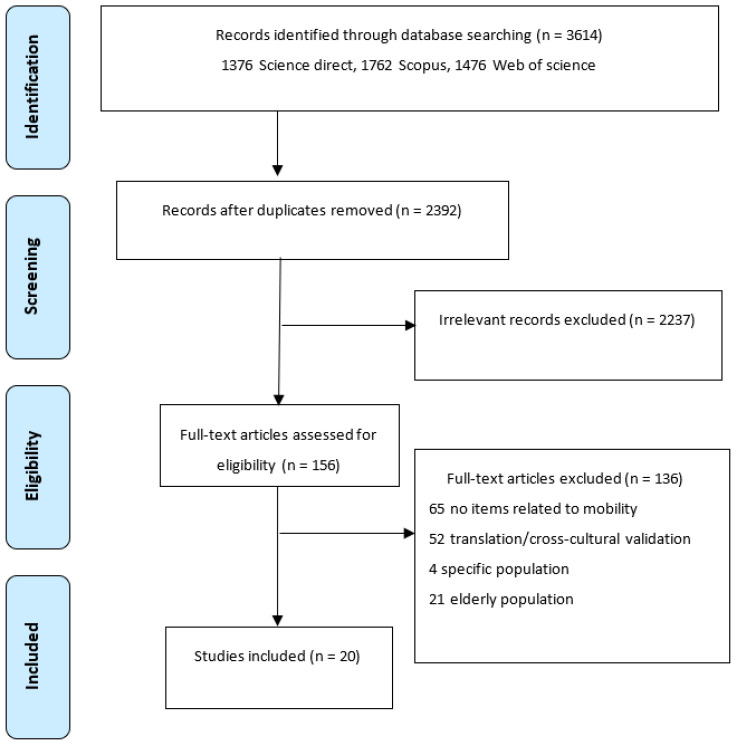
PRISMA flow chart.

**Table 1 ijerph-19-16493-t001:** Search Strategy.

Database	Search Strategy
Science Direct	Terms: mobility OR function OR physical; Title, abstract or author-specified keywords: “health-related quality of life” AND (psychometrics OR validity OR reliability) AND NOT (children OR elderly OR adolescents); Title: scale OR questionnaire OR survey OR test OR index.
Scopus	TITLE-ABS-KEY (“health-related quality of life”) AND TITLE (scale OR questionnaire OR survey OR test OR instrument OR index) AND TITLE-ABS-KEY ((psychometrics OR valid * OR reliability) AND NOT (child * OR elderly OR pediatric)) AND ALL (mobility OR function * OR physical)
Web of Science (includes Medline, Current Contents Connect, Derwent Innovations Index, KCI-Korean Journal Database, Russian Science Citation Index, SciELO Citation Index, and the main Web of Science collection)	TS = “health-related quality of life” AND TI = (scale OR questionnaire OR survey OR test OR instrument OR index) AND TS = (psychometrics OR valid * OR reliability) NOT TS = (child * OR elderly OR pediatric) AND TS = (mobility OR function * OR physical)

**Table 2 ijerph-19-16493-t002:** Characteristics of the general HRQOL instruments included in the review.

PROM	Original Author	Year	Dimensions	Items (Mobility-Related Items)	Answer Choices
SF-36 [47]	Ware	1992	Physical functioning	36 (10)	1 limited a lot—3 not limited at all
	Sherbourne		Role physical		1 all of the time—5 none of the time
			Social functioning		1 all of the time/not at all—5 none of the time/extremely
			Bodily pain		1 lowest—5/6 highest level of pain
			General mental health		1 all of the time—5 none of the time
			Role emotional		1 all of the time—5 none of the time
			Vitality		1 all of the time—5 none of the time
			General health		1 excellent or definitely true—5 poor or definitely false
SF-12 [33]	Ware et al.	1996	Physical functioning	12 (2)	1 limited a lot—3 not limited at all
			Role-Physical		1 all of the time—5 none of the time
			Bodily pain		1 not at all—5 extremely
			General health		1 excellent-5 poor
			Vitality		1 all of the time—5 none of the time
			Social functioning		1 all of the time—5 none of the time
			Role-Emotional		1 all of the time—5 none of the time
			Mental health		1 all of the time—5 none of the time
SF-6D [48]	Brazier et al.	2002	Physical functioning	6 (1)	1 not limit—6 a lot of limit
			Role limitations		1 no problem—4 limited
			Social functioning		1 none of the time—5 all of the time
			Pain		1 no pain—6 extremely
			Mental health		1 none of the time—5 all of the time
			Vitality		1 all of the time—5 none of the time
AQoL [29]	Hawthorne, et al.	1999	Ilness	15 (3)	1 highest level—4 lowest level of QOL
			Independent living		
			Social relationships		
			Physical senses		
			Pshycological well-being		
AQoL-8 [27]	Hawthorne	2009	Independent living	8 (2)	1 highest level—4 lowest level of QOL
			Social relationships		
			Physical senses		
			Pshycological well-being		
AQoL-6D [28]	Richardson, et al.	2012	Independent living	20 (4)	4–6 (depending on the question where it goes from the highest QOL level to the lowest)
			Relationships		
			Mental health		
			Coping		
			Pain		
			Senses		
EQ-5D [49]	EuroQol group	1990	Mobility	6 (1)	3 and VAS (0–100).
(EQ-5D-5L [38])		(2011)	Self-care		(5 and VAS (0–100))
			Usual Activities		
			Pain/Discomfort		
			Anxiety/Depression		
WHOQOL- BREF [50]	WHOQOL group	1998	Physical health	26 (1)	5
			Psychological		
			Social relationships		
			Environment		
HUI-2 [51]	Feeny et al.	1992	Sensation	7 (1)	3–5
			Mobility		
			Emotion		
			Cognition		
			Self-care		
			Pain		
			Fertility *		
HUI-3 [52]	Feeny et al.	1995	Vision	8 (2)	5–6
			Hearing		
			Speech		
			Ambulation		
			Dexterity		
			Emotion		
			Cognition		
			Pain		
NHP [53]	Hunt McEwen	1980	Physical mobility	38 (8)	2 (yes–no)
			Energy		
			Sleep		
			Pain		
			Social isolation		
			Emotional reactions		
HINT-20 [44]	Jo et al.	2017	Physical health	20 (3)	4
			Social health		
			Mental health		
			Positive health		
QWB-SA [54]	Kaplan et al.	1997	Mobility	76 (11)	2 (yes–no questions)
			Physical activity		5 (multiple stage questions)
			Social activity		4 (3-day recall questions)
			58 symptom/problem complexes		
Stark QoL [46]	Hardt	2015	Mood	9 (6)	Mood and physical functioning: 5
			Energy	16 pictures	Energy: 2
			Social Contact		Social contact: 3
			Physical functioning		
PAT-5D-QOL [45]	Kopec et al.	2013	Walking	30 (18)	4
			Handling objects		
			Daily activities		
			Pain or discomfort		
			Feelings		
HALex [43]	Erickson	1998	Health perceived	11 (1)	General health: 5
			Activity limitation		Activity limitation: 2 (yes–no) and one question 4 answer choices.
FSQ [55]	Jette et al.	1986	Physical function	34 (9)	4 no difficulty—1 usually did not do because of health, 0 usually did not do for other reasons
			Pshycological function		1 all of the time—6 none of the time
			Social function		1 all of the time—6 none of the time
			Role function		1 all of the time—6 none of the time
SWED-QUAL [32]	Brorsson, et al.	1993	Physical functioning	61 (9)	4–5
			Pain		
			Role functioning		
			Emotional well-being		
			Sleep		
			General health		
			Family functioning		
MQLI [56]	Mezzich, et al.	2000	Physical well-being	10 (2)	VAS (1–10)
			Psychological/Emotional well-being		
			Self-care and Independent Functioning		
			Occupational functioning		
			Interpersonal functioning		
			Social emotional support		
			Community and services support		
			Personal fulfillment		
			Spiritual fulfillment		
			Overall quality of life		
CAT-Health [40]	Rebollo et al.	2009	HRQOL (unidimensional)	96 (25)	5

Abbreviations: HRQOL, health-related quality of life; PROM, patient reported outcome measure; QOL, quality of life; VAS, visual analogue scale; SF-36, Medical Outcome Study Short Form 36; SF-12, Medical Outcome Study Short Form 12; SF-6D, Medical Outcome Study Short Form 6 Dimensions; AQoL, Assessment of Quality of Life; AQoL-8, Assessment of Quality of Life 8; AQoL-6D, Assessment of Quality of Life 6D; EQ-5D, European Quality of Life Instrument (5L, 5 level); WHOQOL-BREF, World Health Organization Quality of life; HUI, Health Utility Index; NHP, Nottingham Health Profile; HINT-20, Health-related Quality of Life Instrument with 20 items; QWB-SA, Quality of Well-being Self-Administered; PAT-5D-QOL, Paper-and-pencil semi-adaptive questionnaire for 5 domains of health-related quality of life; HALex, health and activity limitation index; FSQ, functional status questionnaire; SWED-QUAL, Swedish Health-related Quality of Life Survey; MQLI, multicultural quality of life index; CAT, computer-adaptive test. * Fertility was included because the original application was concerned about sub-fertility and infertility sequelae associated with childhood cancer and its treatment. Fertility is not assessed using current HUI questionnaires.

**Table 3 ijerph-19-16493-t003:** Characteristics of the studies assessing the measurement properties of general HRQOL instruments.

PROM		Population			Instrument Administration		
	Author Year	Sample Size	Age	Gender	Setting	Country	Language
			Mean (SD, Range) Year	% Female			
AQoL	Hawthorne 2001	976	52.4 (18)	50	nr	Australia	English
AQoL	Hawthorne 1999	255	nr	53	nr	Australia	English
AQoL-6D	Richardson 2012	620	nr	53.50	nr	Australia	English
AQoL-8	Hawthorne 2009	3015	45 (19)	51	nr	Australia	English
CAT-Health	Rebollo 2009	Pilot study = 185 1373	Pilot study = 50.97 (22.74) 50.03 (18.22)		SA (online)	Spain	Spanish
CAT-Health	Rebollo 2010	396	48.6 (17.7)	67	SA (online)	Spain	Spanish
EQ-5D	Johnson 2000	1518	53.51 (16.61)	33.5	SA (online)	Canada	English
EQ-5D	Van Agt 1994	208	49.3 (18.1)	43.3	SA	Netherlands	Dutch
EQ-5D/HUI-2/HUI-3/SF-6D/QWB-SA	Palta 2011	3844 general population/265 cataract patients	35–89	57.3	Telephone interviewer (general population)/SA (cataract patients)	USA	English
EQ-5D-5L	Herdman 2011	144	nr	nr	nr	UK y España	English and Spanish
FSQ	Cleary 2000	9267 (from several samples)	nr	nr	nr	USA	English
HALex	Erickson 1998	41,104	43.86 (0.15)	52.59	Telephone administered	USA	English
HINT-20	Jo 2017	1191	47 (14.8) 44.6 (13.2) 45.1 (13.3)	53.4 50.3 52	SA	Korea	Korean and English
HUI-III	Boyle 1995	506	nr	nr	nr	Canada	English/French
MQLI	Schwartz 2006	260	nr	nr	Interviewer administered	Peru	Spanish
NHP/SF-36	Krantz 2019	412	62.8 (range 39–78)	77	SA	Sweden	English
PAT-5D-QOL	Kopec 2013	1349 (version 1)	67	60	SA	Canada	English
SF-12	Ware 1996	3363	nr	nr	SA	USA	English
StarkQoL	Hardt 2015	500	44.82 (16.11)	50	SA (online)	Germany	German/English
SWED-QUAL	Brorsson 1993	1143/1396	nr	nr	SA (postal)	Sweden	Swedish/English
WHOQOL-Bref	Hawthorne 2006	931	48.2 (17.3)	54	SA	Australia	English

Abbreviations: nr, not reported; PROM, patient-reported outcome measure; SA, self-administered; USA, United States of America; SF-36, Medical Outcome Study Short Form 36; SF-12, Medical Outcome Study Short Form 12; SF-6D, Medical Outcome Study Short Form 6 Dimensions; AQoL, Assessment of Quality of Life; AQoL-8, Assessment of Quality of Life 8; AQoL-6D, Assessment of Quality of Life 6D; EQ-5D, European Quality of Life Instrument (5L, 5 level); WHOQOL-BREF, World Health Organization Quality of life; HUI, Health Utility Index; NHP, Nottingham Health Profile; HINT-20, Health-related Quality of Life Instrument with 20 items; QWB-SA, Quality of Well-being Self-Administered; PAT-5D-QOL, Paper-and-pencil semi-adaptive questionnaire for 5 domains of health-related quality of life; HALex, health and activity limitation index; FSQ, functional status questionnaire; SWED-QUAL, Swedish Health-related Quality of Life Survey; MQLI, multicultural quality of life index; CAT, computer-adaptive test.

**Table 4 ijerph-19-16493-t004:** Body functions categories ICF linking of HRQOL instruments.

ICF Category	HUI-2	QWB-SA	PAT-5D-QoL	SWED-QUAL	CAT-Health
b152 Emotional functions		1			
b455 Exercise tolerance functions				1	
b760 Control of voluntary movement functions			1		1
b7603 Supportive functions of arm or leg	1				
b770 Gait pattern functions		1			

Abbreviations: ICF, International Classification of Functioning, Disability and Health; HUI, Health Utility Index; QWB-SA, Quality of Well-being Self-Administered; PAT-5D-QOL, Paper-and-pencil semi-adaptive questionnaire for 5 domains of health-related quality of life; SWED-QUAL, Swedish Health-related Quality of Life Survey; CAT, computer-adaptive test.

**Table 5 ijerph-19-16493-t005:** Environmental factors categories ICF linking of HRQOL instruments.

ICF Category	AQoL	AQoL 8	HUI-3	NHP
e1151 Assistive products and technology for personal use in daily living			1	
e1250 General products and technology for communication	3	2		
e3 Support and relationships			2	1

Abbreviations: ICF, International Classification of Functioning, Disability and Health; AQoL, Assessment of Quality of Life; AQoL-8, Assessment of Quality of Life 8; HUI, Health Utility Index; NHP, Nottingham Health Profile.

**Table 6 ijerph-19-16493-t006:** Activities and participation categories two level ICF linking of HRQOL instruments.

ICF Category	SF-36	SF-12	SF-6D	AQoL	AQoL 8	AQoL-6D	EQ 5D	EQ 5D (5L)	WHOQOL-Bref	Stark Qol	HUI-2	HUI-3	NHP	HINT-20	QWB-SA	PAT-5D-QoL	HALex	FSQ	SWED-QUAL	MQLI	CAT-Health
d Activities and participation	2	1	4												1	2		1	2		1
d177 Making decisions																				1	
d230 Carrying out daily routine																2				1	1
d4 Mobility							1	1							1						1
d410 Changing basic body position	2									1	1		1		2	1		2	1		5
d415 Maintaining a body position													2		3	5					2
d430 Lifting and carrying objects	2									3	1				1	2		1	1		1
d440 Fine hand use												1				4			1		1
d445 Hand and arm use	1	1									1		1			13					
d449 Carrying, moving and handling objects, other specified and unspecified	1	1								1						1					
d450 Walking	3						1	1			2	4	3	1	1	9		3	1		1
d455 Moving around	3	1				1			1		3	1	1	1	1	5		2	1		4
d460 Moving around in different locations																1		1			1
d465 Moving around using equipment						1					1	1	1		1						
d469 Walking and moving, other specified and unspecified				1	1	1															
d470 Using transportation																		1			
d475 Driving																		1			
d498 Mobility, other specified							1														
d5 Self-care				1														1			
d510 Washing oneself	1		2			1										1		1			1
d520 Caring for body parts						1															
d530 Toileting						1															
d540 Dressing	1		2			1				1			1			2		1	1		2
d550 Eating						1										2		1			
d6 Domestic life																	1	1		1	
d620 Acquisition of goods and services						1									1	3		1			
d630 Preparing meals				1	1	1															1
d640 Doing housework						1							1	1	1	8		1			
d649 Household tasks, other specified and unspecified				1	1	1															
d650 Caring for household objects				2	2	1										2		2	1		
d750 Informal social relationships																3					
d760 Family relationships																3					
d820 School education															1	2	1			1	
d850 Remunerative employment															1	3	1			1	
d920 Recreation and leisure	1	2				1									3	2		1	2		2
d930 Religion and spirituality															1						

Abbreviations: ICF, International Classification of Functioning, Disability and Health; SF-36, Medical Outcome Study Short Form 36; SF-12, Medical Outcome Study Short Form 12; SF-6D, Medical Outcome Study Short Form 6 Dimensions; AQoL, Assessment of Quality of Life; AQoL-8, Assessment of Quality of Life 8; AQoL-6D, Assessment of Quality of Life 6D; EQ-5D, European Quality of Life Instrument (5L, 5 level); WHOQOL-BREF, World Health Organization Quality of life; HUI, Health Utility Index; NHP, Nottingham Health Profile; HINT-20, Health-related Quality of Life Instrument with 20 items; QWB-SA, Quality of Well-being Self-Administered; PAT-5D-QOL, Paper-and-pencil semi-adaptive questionnaire for 5 domains of health-related quality of life; HALex, health and activity limitation index; FSQ, functional status questionnaire; SWED-QUAL, Swedish Health-related Quality of Life Survey; MQLI, multicultural quality of life index; CAT, computer-adaptive test.

**Table 7 ijerph-19-16493-t007:** Quality of PROM development and concept elicitation studies.

PROM	PROM Design							Total PROM Development
	General Design Requirements					Concept Elicitation	Total PROM Design	
	Clear Construct	Clear Origin of Construct	Clear Target Population for Which the PROM Was Developed	Clear Context of Use	PROM Developed in Sample Representing the Target Population			
SF-12 [33,57]	V	V	V	V	I		I	I
AQOL [30,58]	V	V	V	V	I		I	I
SF-36 [57]	V	V	V	V	I			I
NHP [57]	D	D	I	D	I		I	I
EQ-5D [37,57]	I	D	V	V	I		I	I
CAT-Health [40]	V	V	V	D	D	I	I	I
EQ-5D-5L [38,58]	V	V	V	V	D	I	I	I
HINT-20 [44]	V	V	V	V	V	D	D	I
HUI 2 and 3 [58]	V	V	V	V	I		I	I

Ratings: V, very good; A, adequate; D, doubtful; I, inadequate. When the PROM was not developed in a sample representing the target population, the concept elicitation was not further rated. Abbreviations: SF-36, Medical Outcome Study Short Form 36; SF-12, Medical Outcome Study Short Form 12; AQoL, Assessment of Quality of Life; EQ-5D-5L, European Quality of Life Instrument 5 level; HUI, Health Utility Index; NHP, Nottingham Health Profile; HINT-20, Health-related Quality of Life Instrument with 20 items; CAT, computer-adaptive test.

**Table 8 ijerph-19-16493-t008:** Evidence synthesis on measurement properties of HRQOL instruments in general population.

PROM	Structural Validity		Internal Consistency		Reliability		Measurement Error		Criterion Validity		Hypotheses Testing for Construct Validity		Responsiveness	
	Meth Quality	Rating	Meth Quality	Rating	Meth Quality	Rating	Meth Quality	Rating	Meth Quality	Rating	Meth Quality	Rating	Meth Quality	Rating
SF-12	Very low	?			Moderate	+					Low	?		
Aqol	Moderate	-	High	-							High	?	Moderate	?
AQoL-8	Moderate	?							High	+	Moderate	?		
AQoL-6D	High	+												
SF-36			High	+							High	+		
SF-6D			High	+	Very low	-	Low	?						
NHP			High	-							High	+		
Stark QoL	Low	+	High	-							High	?		
EQ-5D			High	+	Very low	-	Low	?	High	-				
CAT-Health	High	?	High	+							High	+		
HINT-20	Moderate	-	High	+	Low	-					Moderate	+		
HALex											Moderate			
FSQ			High	-					Very low	-	Moderate	?		
HUI-3			High	+	Very low	+	Low	?						
HUI-2			High	+	Very low	-	Low	?						
QWB			High	-	Very low	-	Low	?						
SWED-QUAL			High	+							Low			
MQLI	Very low	?	Low	?	Very low	-					Low			
WHOQOL-BREF			High	-							Moderate	?		

Rating: +, sufficient; -, insufficient; ?, indeterminate. Abbreviations: SF-36, Medical Outcome Study Short Form 36; SF-12, Medical Outcome Study Short Form 312; SF-6D, Medical Outcome Study Short Form 6 Dimensions; AQoL, Assessment of Quality of Life; AQoL-8, Assessment of Quality of Life 8; AQoL-6D, Assessment of Quality of Life 6D; EQ-5D, European Quality of Life Instrument; WHOQOL-BREF, World Health Organization Quality of life; HUI, Health Utility Index; NHP, Nottingham Health Profile; HINT-20, Health-related Quality of Life Instrument with 20 items; QWB-SA, Quality of Well-being Self-Administered; PAT-5D-QOL, Paper-and-pencil semi-adaptive questionnaire for 5 domains of health-related quality of life; HALex, health and activity limitation index; FSQ, functional status questionnaire; SWED-QUAL, Swedish Health-related Quality of Life Survey; MQLI, multicultural quality of life index; CAT, computer-adaptive test.

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
