# Peer review of "Health-Related Quality of Life (HRQOL) Instruments and Mobility: A Systematic Review"

_ijerph, 2022, doi:10.3390/ijerph192416493_

Round 1

Reviewer 1 Report

Conclusively, I suggest accepting this article after a minor revision. This study reviewed those papers relevant to the mobility covered by the health-related quality of life (HRQOL) questionnaires based on the International Classification of Functioning, Disability and Health (ICF) and to evaluate their quality. Nevertheless, I think readers of this manuscript would not like to read such contents. They would like to read the authors' opinions after the review. For example, the authors may discuss the further research to their interested topics of this study. Please add such contents.

Author Response

I would like to thank you for the comments received and I sincerely apologize for the mistakes made.

Both the introduction and the discussion have been modified in order to present a more in-depth analysis with a greater contribution of ideas on how to apply these questionnaires in the clinical practice and what could be relevant lines of research to develop in this area, especially the improvement in the development and evaluation of the quality and psychometric properties of the questionnaires.

All changes made to the manuscript have been highlighted.

I am at your disposal for any additional comment.

Reviewer 2 Report

This is an interesting study which aims to identify generic HRQoL measures that include a domain related to mobility or physical function and assess their quality as well as to compare the content of the domains related to mobility covered by the HRQoL questionnaires based on the International Classification of Functioning, Disability and Health.

I have several points which I hope will be useful for the authors:

1.      The introduction should state the research gap and why there is a need to extend the existing literature.

2.      The study should highlight its contribution to the field.

3.      In introduction the study states “Therefore, most generic HRQOL instruments cover all three domains”. Clearly define for the reader “generic HRQoL”.

4.      Discussion in its present form is difficult to follow. I suggest streamlining the key points, identifying clearly the strengths and limitations of the study, and identifying priorities for future research.

5.      The study would benefit from improving the English. For example “patient perspective is especially important in research and health-care provide” do you mean provision of healthcare? There are grammar mistakes and the style should be improved. 

Author Response

I have several points which I hope will be useful for the authors:

I would like to thank you for the comments received and I sincerely apologize for the mistakes made. I shall try to reply specifically to all points.

  1. The introduction should state the research gap and why there is a need to extend the existing literature. We have included a paragraph (lines 64-69) in the introduction that includes the existing literature and the gap that we have tried to fill.
  2. The study should highlight its contribution to the field. We have added changes to both the introduction and discussion and emphasized our contribution. All changes have been highlighted.
  3. In introduction the study states “Therefore, most generic HRQOL instruments cover all three domains”. Clearly define for the reader “generic HRQoL”. Thank you for your comment. We have included the definition in the manuscript on line 50.
  4. Discussion in its present form is difficult to follow. I suggest streamlining the key points, identifying clearly the strengths and limitations of the study, and identifying priorities for future research. Thanks for the suggestion. We have modified the discussion and added a paragraph on strengths (lines 291-294). We have also included the need to develop new instruments for the future with adequate quality.
  5. The study would benefit from improving the English. For example “patient perspective is especially important in research and health-care provide” do you mean provision of healthcare? There are grammar mistakes and the style should be improved. Thanks for your suggestion. We apologize for the poor language of our manuscript. We have now worked on both language and readability and have also involved native English speakers for language corrections. We really hope that the language level has been substantially improved.

Finally, I would like to thank you again for all your comments and I hope I have clarified the questions. All changes made to the manuscript have been highlighted.

I am at your disposal for any additional comment.

Round 2

Reviewer 2 Report

Thank you for your additional work on the manuscript and for addressing my suggestions. I have couple more suggestions which might improve the manuscript.

In Abstract I think the following flow of statement makes sense "The objective of this work was compare the content of the domains related to mobility covered by the health-related quality of life (HRQOL) questionnaires based on the International Classification of Functioning, Disability and Health (ICF) and to evaluate their quality.  The quality assessment was performed according to the COnsensus-based Standards for the selection of health Measurement INstruments (COSMIN) guidance.  For this, a systematic review was carried out in the databases Scopus, Web of Science and Science Direct. " because you define "quality" after it is mentioned the first time. 

  • Specify the date of inception of the search.
  • State in the limitation that not all databases were searched such as PsycINFO; EconLit;
  • State that psychometric properties evaluated were not drawn from all possible sources such as: Smith S, Lamping D, Banerjee S, Harwood R, Foley B, Smith P, et al. Measurement of health-related quality of life for people with dementia: development of a new instrument (DEMQOL) and an evaluation of current methodology. Health Technology Assessment (Winchester, England). 2005;9(10):1-iv.; Food and Drug Administration. Patient reported outcome measures: use in medical product development to support labelling claims. Washington DC. 2009.; Lohr KN. Assessing health status and quality-of-life instruments: attributes and review criteria. Quality of life Research. 2002;11(3):193-205. Thus, not all psychometric properties were assessed.
  • I think it is important each standard that was rated is defined in the text, as well as the definition for each four-point rating is also defined.

Author Response

Thank you for your additional work on the manuscript and for addressing my suggestions. I have couple more suggestions which might improve the manuscript.

I would like to thank you for the comments received. I shall try to reply specifically to all the commentaries.

  1. In Abstract I think the following flow of statement makes sense "The objective of this work was compare the content of the domains related to mobility covered by the health-related quality of life (HRQOL) questionnaires based on the International Classification of Functioning, Disability and Health (ICF) and to evaluate their quality.  The quality assessment was performed according to the COnsensus-based Standards for the selection of health Measurement INstruments (COSMIN) guidance.  For this, a systematic review was carried out in the databases Scopus, Web of Science and Science Direct. " because you define "quality" after it is mentioned the first time. 

Thank you very much for this suggestion. We have modified the sentences in the introduction.

  1. Specify the date of inception of the search.

The search was during the whole month of march. We have put it correctly on line 79.

  1. State in the limitation that not all databases were searched such as PsycINFO; EconLit;

Thank you for your comment. We have added this limitation in the manuscript on line 288.

  1. State that psychometric properties evaluated were not drawn from all possible sources such as: Smith S, Lamping D, Banerjee S, Harwood R, Foley B, Smith P, et al. Measurement of health-related quality of life for people with dementia: development of a new instrument (DEMQOL) and an evaluation of current methodology. Health Technology Assessment (Winchester, England). 2005;9(10):1-iv.; Food and Drug Administration. Patient reported outcome measures: use in medical product development to support labelling claims. Washington DC. 2009.; Lohr KN. Assessing health status and quality-of-life instruments: attributes and review criteria. Quality of life Research. 2002;11(3):193-205. Thus, not all psychometric properties were assessed.

On line 88 we stated that the studies were restricted if the validation of the instrument was in a population with a specific pathology. On line 119 we included the measurement properties assessed: “content validity, structural validity, internal consistency, reliability, measurement error, criterion validity and construct validity”. Moreover, we have made changes in the quality assessment section to clarify. Lastly, we have added a limitation in the discussion (line 298-301)

  1. I think it is important each standard that was rated is defined in the text, as well as the definition for each four-point rating is also defined.

Thank for your comment. We have added the overall definition ratings of each measurement property (i.e., sufficient (+), insufficient (–), inconsistent (±)). Moreover, a concept of a grading for the quality (GRADE approach) has also included. On the other hand, we have included the references where the requirements for the awarding of each four-point rating ( Mokkink, L. B.,  Prinsen, C. A., Patrick, D. L., Alonso, J., Bouter, L. M., de Vet, H. C., & Terwee, C. B. (2018). COSMIN methodology for systematic reviews of Patient-Reported Outcome Measures (PROMs) – user manual. http://www.cosmin.nl/) and the criteria for good measurement properties (Prinsen, C.A.C.; Mokkink, L.B.; Bouter, L.M.; Alonso, J.; Patrick, D.L.; de Vet, H.C.W.; Terwee, C.B. COSMIN Guideline for Systematic Reviews of Patient-Reported Outcome Measures. Qual. Life Res. 2018, 27, 1147–1157, doi:10.1007/s11136-018-1798-3).

Finally, I would like to thank you again for all your comments and I hope I have clarified the questions. All changes made to the manuscript have been highlighted.

I am at your disposal for any additional comment.